# Factors associated with anaemia in pregnancy: A retrospective cross-sectional study in the Bolgatanga Municipality, northern Ghana

**Donatus Nbonibe Abaane**[1,2]***, Martin Nyaaba Adokiya**[3]**, Gilbert Abotisem Abiiro**[4,5]

**1** Department of Global and International Health, School of Public Health, University for Development Studies, Tamale, Ghana, **2** District Nutrition Unit, District Health Directorate, Ghana Health Service, Garu, Ghana, **3** Department of Epidemiology, Biostatistics and Disease Control, School of Public Health, University for Development Studies, Tamale, Ghana, **4** Department of Health Services, Policy, Planning, Management and Economics, School of Public Health, University for Development Studies, Tamale, Ghana, **5** Department of Population and Reproductive Health, School of Public Health, University for Development Studies, Tamale, Ghana

* abaanedonatus1@gmail.com

**Data Availability Statement:** The dataset is publicly available at: https://doi.org/10.5061/dryad.rbnzs7hcq.

## Abstract

### Background

Anaemia in pregnancy (AIP) remains a severe public health problem associated with adverse outcomes. This study assessed haemoglobin levels and the prevalence of anaemia during antenatal care (ANC) registration, at 28 weeks and 36 weeks of gestation as well as the factors associated with AIP at the different stages of pregnancy.

### Methods

A retrospective cross-sectional design was implemented. Using ANC registers as the sampling frame, 372 pregnant women, within 36 and 40 weeks of gestation were randomly sampled from 28 health facilities for the study. The participants were all receiving ANC in the Bolgatanga Municipality. Data were collected via clinical records review and a questionnaire-based survey between October and November, 2020. Using the Statistical Package for the Social Sciences (SPSS), descriptive analysis of haemoglobin levels and the prevalence of anaemia were performed. In addition, binary logistic regression was used to identify the factors associated with anaemia in pregnancy. AIP was determined using the national practice of 11.0g/dl haemoglobin cut-off point and the World Health Organisation's recommended adjustment for the 2nd trimester of pregnancy was made using the cut-off of 10.5g/dl to account for the effect of haemodilution.

### Results

At booking, AIP prevalence was 35.8% (95%CI:30.9, 40.9) using a cut-off of 11.0g/dl and 25.3% (95%CI:20.9, 30.0) using a cut-off of 10.5g/dl for those in the 2nd trimester. At 28 weeks, AIP prevalence was 53.1% (95%CI:45.8, 60.3) and 37.5 (95%CI:30.6, 44.8) using a

**Funding:** The author(s) received no specific funding for this work.

**Competing interests:** The authors have declared that no competing interests exist.

**Abbreviations:** AIP, Anaemia in pregnancy; ANC, Antenatal care; AOR, Adjusted odd ratio; CHPS, Community-based Health Planning and Services; CI, Confidence interval; GHS, Ghana Health Services; GSS, Ghana Statistical Service; IFA, Iron and Folic Acid; NHIS, National Health Insurance Scheme; WHO, World Health Organization.

cut-off of 11.0g/dl and 10.5g/dl for those in the 2$^{nd}$ trimester, respectively. At 36 weeks, AIP prevalence was 44.8% (95%CI:39.2, 50.4) using a cut-off of 11.0g/dl. At p<0.05, registering after the first trimester (AOR = 1.87, 95%CI: 1.17, 2.98, P = 0.009) and at a regional hospital (AOR = 2.25, 95%CI: 1.02, 4.98, P = 0.044) were associated with increased odds of AIP but registering at a private hospital (AOR = 0.32, 95%CI: 0.11, 0.92, P = 0.035) was associated with decreased odds of AIP at booking. At 28 weeks, age group 26–35 years (AOR = 0.46, 95%CI: 0.21, 0.98, P = 0.044), Christianity (AOR = 0.32, 95%CI: 0.31, 0.89, P = 0.028.), high wealth (AOR = 0.27, 95%CI: 0.09, 0.83, P = 0.022) and tertiary education (AOR = 0.09, 95%CI:0.02, 0.54, P = 0.009) were associated with decreased odds of AIP. At 36 weeks, booking after first trimester of pregnancy was associated with increased odds (AOR = 1.72, 95%CI: 1.05, 2.84, P = 0.033) whilst high wealth (AOR = 0.44, 95%CI: 0.20, 0.99, P = 0.049), higher age groups—26–35 (AOR = 0.38, 95%CI: 0.21, 0.68, P = 0.001) and 36–49 years (AOR = 0.35, 95%CI: 0.13, 0.90, P = 0.024) and secondary education of spouse were associated with reduced odds (AOR = 0.35, 95%CI: 0.14, 0.88, P = 0.026) of AIP.

## Conclusion

AIP consistently increased from registration to 36 weeks of gestation. Given the observed correlates of AIP, we recommend that interventions geared towards early ANC registration, improved household wealth, and improved maternal education are required to reduce AIP.

## Introduction

Anaemia in pregnancy (AIP) is a pathophysiological state in pregnancy resulting from a woman's haemoglobin level falling below 11.0g/dl [1, 2]. The World Health Organisation (WHO) further recommends a threshold of 10.5g/dl for AIP diagnosis in the second trimester of pregnancy, which is the peak of haemodilution [3–5]. However, Ghana uses 11.0g/dl to diagnose AIP in all pregnant women across the three trimesters of pregnancy [2]. AIP is diagnosed by red blood cell counts using Automatic Haemoglobin Analyzer or photometric devices [1, 6, 7]. Anaemia impairs the functioning of every human organ including the reproductive system [6, 8–10]. AIP has been linked to several undesirable pregnancy-related outcomes such as intra-uterine growth restriction, poor "Appearance, Pulse, Grimace, Activity, and Respiration" score, preterm birth, stillbirths, postpartum haemorrhage, abortion, low birthweight, perinatal and maternal mortalities [2, 6, 11–16]. Pregnancy increases vulnerability to anaemia due to induced physiological changes [6, 15, 17–22]. While poor production or abnormal loss of red blood cells during pregnancy [23] is the immediate cause of AIP, this is often triggered by a preceding hierarchy of factors including malnutrition and food insecurity, pre-existing disease conditions, insanitary and unhealthy environment, poor individual or household level socio-economic characteristics, particularly, low education, unemployment and poor access to appropriate healthcare.

ANC has been established as an effective intervention of controlling AIP [4]. It is composed of a set of interventions that reduces the effect of the underlying causes of AIP. These are iron folate supplementation, infection prevention, identification and management of non-communicable diseases, early detection and management as well as referral to higher care level for effective management of AIP [4]. ANC also involves social and behavioural change communication activities. Some of the activities are health education, counselling, advocacy, and

community mobilization meant to prevent AIP [2, 4, 24]. During ANC, three mandatory haemoglobin level checks are done, at registration, at 28 weeks and at 36 weeks of gestation to enable the detection, prevention and treatment of anaemia [4]. Once AIP is diagnosed and treatment commences, bi-weekly haemoglobin checks are recommended for monitoring progress [25]. With expansion of geographical service coverage, through the Community-based Health Planning and Services (CHPS) model, about 98% of pregnant women attend ANC and about 89% had at least four ANC visits during pregnancy in Ghana [26].

Despite the increased ANC attendance and the comprehensive AIP control interventions which are provided during ANC, studies have shown that the global prevalence of AIP remains about 40.1% [6, 27, 28]. Asia has the highest burden of AIP (48.7%) and Africa being the second with a prevalence of 46.3% [29]. The AIP prevalence in sub-Saharan Africa is estimated between 44% and 48% [30, 31]. The Ghana Demographic and Health Survey in 2014 and the Ghana Micronutrients Survey in 2017 reported the AIP prevalence as 45% in Ghana [32, 33]. An earlier cross-sectional study reported a relatively higher AIP prevalence of 50.4% in the Bolgatanga Municipality of Ghana [34]. These studies have mainly assessed the point prevalence and determinants of AIP in Ghana and beyond [34–44]. However, few studies conducted elsewhere [45, 46] assessed changes in haemoglobin levels and anaemia status of women in the course of pregnancy. In Ghana, Adu-Afarwuah et al, assessed the impact of iron supplementation on haemoglobin levels of pregnant women using a randomised controlled trial involving haemoglobin checks at two points of gestation [47]. We could not identify studies in Ghana assessing haemoglobin levels and anaemia status among pregnant women at all the three WHO recommended points for haemoglobin check during ANC and their determinants in the course of gestation. Some studies assessed AIP at one point of gestation [35, 48] and others at two points [49, 50], but no study assessed the phenomenon at all the three recommended haemoglobin check points during ANC. These three points are used for decision making, hence, knowledge on anaemia prevalence and its determinants at the three stages of pregnancy would be useful in the care process. This study, therefore, assessed haemoglobin levels and the prevalence of anaemia at ANC registration, at 28 weeks and 36 weeks of gestation and the socio-economic factors associated with AIP at these stages of pregnancy.

## Methods

### Study setting

The study was conducted in health facilities providing ANC in the Bolgatanga Municipality of Ghana. The Bolgatanga Municipality has an estimated population of 130,890. Females are 67,062 (51.2%) of which 31,414 (24.0%) are women in the reproductive age of 15–49. With a total fertility rate of 3.0 per woman, about 5,236 pregnancies were expected in 2020 [51]. The Municipality has 41 health facilities located in its nine sub-municipalities. These comprises one regional hospital, seven private health facilities, six public health centres, six clinics, 24 CHPS compounds and one maternity home [51]. Out of this, 28 of the facilities were providing ANC services in 2020 and 18 had laboratories for running the recommended ANC tests including haemoglobin check-ups. The remaining health facilities used portable devices and test kits for haemoglobin checking or referred their clients to other facilities for haemoglobin check-up. Coverage of ANC in 2019 was 93% with a three-year average of 91% in the Municipality [51]. Nearly three-quarters (72.4%) made a 4th ANC visit before delivery [51]. In 2019, haemoglobin checked at registration was near universal (97%) but reduced to about 52% at 36 weeks of gestation [51].

## Study design and sampling

The study employed a quantitative retrospective cross-sectional design. All women in 2020 who registered for ANC before 28 weeks and had reached 36 weeks of gestation during the period of data collection were eligible to be included in the study. Pregnant women with documented or reported genetic causes of anaemia and had experienced severe bleeding were excluded. All participants were randomly sampled at or after 36 weeks of gestation.

With a target population of 4,765 expected ANC registrants in 2020, a confidence interval of 95% and a margin of error of 0.05, the sample size (n) for the study was estimated as 369 using the Yamane Taro's formula [52] given as:

$$n = \frac{N}{1 + Ne^2} = \frac{4765}{1 + 4765 \times 0.05 \times 0.05} = 369$$

Approximately 11.5% was added to cater for potential non-response, resulting in a total sample of 411.

A proportionate sample, based on the number of ANC registrants per facility was drawn from each of the 28 facilities offering ANC in the Municipality. At the facility level, simple random sampling using random numbers was used to select the number of respondents allotted to each facility. The ANC registers of the various facilities constituted the sampling frame.

## Recruitment of participants and data collection

Data were collected via clinical records review and a questionnaire-based survey between October 21, 2020 and November 19, 2020. Data on the timing of ANC registration, number of ANC visits, haemoglobin check-ups, and other socio-demographic characteristics of the sampled respondents were extracted from the maternal and child health record book and laboratory results slips using a pre-prepared checklist. These were indicators of care often recorded in the ANC register. On the other hand, a structured questionnaire was used to collect data on household food security and additional sociodemographic characteristics including household assets, that were not captured in the medical records. Table 1 presents a description of the key variables that were covered by our data collection.

Both the checklist and the questionnaire were pretested. This enabled the researchers to review, identify gaps and refine the tools. The final tools were digitalized using the Kobo Collect App. Four data collectors were trained for a face-to-face administration of the questionnaire and the filling of the checklist. All research assistants were either diploma or bachelor's degree holders. The majority of the data collection was done at the premises of the health facilities. It took about 20–30 minutes to administer each questionnaire. The researchers first made contact visits to the health facilities to identify eligible participants who were then sampled for the study. The midwives at the health facilities, who know the schedules of the pregnant women for ANC, assisted the data collectors in the final contact and selection of the participants. The data collectors first extracted the relevant data from the ANC records of each participant and immediately administered the questionnaire to the pregnant woman. The records review and questionnaire administration were, therefore, conducted at the same time. As a retrospective study, the records review and the questionnaire gathered data on issues within the course of the pregnancy. Where the participants failed to attend ANC, efforts were made to visit them at their residences to administer the questionnaire to them.

**Table 1. Description of the key variables used in the data collection.**

| Variable | Type of variable | Definition of variable | Scale of variable | Categories |
|---|---|---|---|---|
| Anaemia in Pregnancy (AIP) | Dependent | Hb less than 11g/dl in all trimesters. Hb of 10.5g/dl for second trimester to adjust for haemodilution | Binary | Anaemic or Non-anaemic (normal) |
| Severity of anaemia | Dependent | Categorization based on the severity of reduced haemoglobin level below the anaemia cut-off point (11g/dL) | | $\geq$ 11g/dL = Normal or no anaemia; 10.9–9.9g/dL = Mild anaemia; 8.9–7g/dL = moderate anaemia; & < 7g/dL = severe anaemic |
| Marital status | Independent | Having customarily or legally bounded partner or devoted partner the respondent is living with who has initiated the process of meeting the legal or customary requirement for marriage | Binary | single or married |
| Type of marriage | Independent | Either monogamous or polygamous marriage | Binary | Monogamous, Polygamous |
| Type of ANC Provider | Independent | The care level of facility the woman receives ANC | Nominal | CHPS, Health Centre, Private /District Hospital, Regional Hospital |
| Religion | Independent | The religious affiliation of the respondents | Nominal | Christianity, Islam & African Traditional Religion |
| Type of toilet | Dependent | The means by which respondent's household dispose of human excreta and their abilities to prevent diseases related anaemia classified as improved when it better prevents diseases otherwise unimproved. | Binary | Improved or unimproved |
| Health status | Independent | Self-rated health status of respondents | Ordinal | Poor & Fair, good, very good and excellent |
| Parity | Independent | The number of deliveries a woman has had | Ordinal | 0, 1, 2, and $\geq$ 3 |
| Age group | Independent | The age category of respondent | Ordinal | 15–25, 26–35 and $\geq$ 36 |
| Place of residence | Independent | Defined as rural, peri-urban or urban | Nominal | rural, peri-urban or urban |
| Occupation group (respondents or spouses) | Independent | Sector of the economy respondent or her spouse operates including students or the unemployed. | Nominal | Unemployed, Formal Sector Worker & Informal Sector Worker |
| Educational status | Independent | Educational level reached by the respondent and spouse | Ordinal | No formal Education, Basic Education, Secondary education, Tertiary education |
| Household Food Security Status | Independent | The levels of anxiety and uncertainty of food supply, quality, (in)sufficiency and frequency of intake of food by household members measured by the Food Insecurity Access scale. | Ordinal | Severe food insecure, |
| | | | | Mild–Moderate insecure and food Secure |
| Household Wealth Index quintile | Independent | Quintile of ranked summary index of principal components analysis of selected 15 household assets (clock, radio, television, telephone, refrigerator, freezer, tablet, TV channel decoder, bed, table, cabinet/cupboard, bank account, toilet facility used, source of drinking water and electricity) | Ordinal | Quintile 1, 2, 3, 4 & 5 |

## Data analysis

All analyses were performed using the Statistical Package for Social Sciences. Univariate analysis of categorical variables such as parity, level of education and occupation was performed using frequencies and percentages. Numerical variables such as haemoglobin levels, food insecurity score, age, etc. were summarised using mean, and standard deviation. A wealth index was constructed using principal component analysis based on data collected on household assets [53]. Binary logistic regression was used to assess the association between the various socio-economic and demographic characteristics of the mothers and their anaemia status at each of the three stages of haemoglobin assessment and the results presented in adjusted odd ratios (AORs) within 95% confidence interval (CI). All predictors were included into the model based on their relevance to explaining AIP as reported by previous studies and their contextual appropriateness.

### Ethics statement

The University for Development Studies Institutional Review Board granted ethical approval for the conduct of this study (date: 21/10/2020). Permission was obtained from the Upper East Regional Health Directorate, the Bolgatanga Municipal Health Directorate and heads of the sampled health facilities. Written informed consent was obtained from all respondents and/ their parents/guardians after they were given information on the purpose of the study and what was required of them as study participants.

## Results

### Socio-economic characteristics of the respondents

Out of the 411 pregnant women sampled for the study, 375 (91.2%) of them responded. However, due to incomplete data from three participants, the sample size was reduced to 372 during data analysis. Table 2 presents the basic characteristics of the participants. The mean and median ages of the participants were both 27. The highest proportion of the women (47.0%) were within the middle reproductive age group of 26–35 years and resided in urban (33.6%) communities. Most (84.1%) of the participants were Christians, married (94.1%), attained only basic education (44.9%) and were unemployed (44.3%). The highest proportion (42.5%) of the participants lived in food secured households and 45.7% registered for ANC at health centres.

### Haemoglobin levels and AIP at the three stages of assessment

Fig 1 presents the distribution of haemoglobin levels of the pregnant women at the three different stages of assessment. Haemoglobin levels were recorded for all the 372 pregnant women at ANC registration, 192 at 28 weeks of gestation and 315 at 36 weeks of gestation. The median haemoglobin level at registration was 11.5g/dl with a mean value of 11.37±1.5g/dl. At 28 weeks of pregnancy, the median haemoglobin level decreased to 10.9g/dl with a mean value of 10.84 ±1.25. A slight rise in haemoglobin levels was observed at 36 weeks with a median of 11.10g/dl and a mean of 11.06±1.26g/dl.

As shown in Table 3, 35.8% (95% CI: 30.9% - 40.9%) of the women were anaemic at ANC registration but this AIP prevalence was reduced to 25.3% (95%CI: 20.9% - 30.0%) after the WHO recommended adjustment for haemodilution for those registering in their second trimester of pregnancy. At 28 weeks of gestation, 53.1% (95% CI: 45.8% - 60.3%) of the pregnant women were anaemic but when the recommended adjustment was made for haemodilution, this prevalence reduced to 37.5% (95% CI: 30.6–44.8). At the 36 weeks of gestation, the AIP prevalence decreased slightly to 44.8% (95% CI: 39.2% - 50.4%). At 36 weeks, haemodilution is resolved to levels that no longer require adjustment.

The severity of AIP at the various stages of gestation is illustrated in Fig 2. At registration, 19.9% of the women had mild and 15.8% had either moderate or severe AIP. After the WHO recommended adjustment for haemodilution, 13.7% of the registrants had mild and 11.6% had moderate or severe anaemia at registration. At 28 weeks of gestation, 32.8% of the women had mild and 20.7% had either moderate or severe anaemia, and at 36 weeks of gestation, 27.0% of the women had mild and 17.8% had moderate or severe anaemia.

### Factors associated with AIP during ANC

Table 4 presents the results from binary logistic regression models on the association between socio-economic characteristics and anaemia status at the three stages of assessment.

From Table 4, women who registered at the Regional Hospital for ANC were more than twice likely to be anaemic compared to those who registered at a CHPS compound

**Table 2. Socio-demographic characteristics of pregnant women (N = 372).**

| Variable | Frequency | Percentage (%) |
|---|---|---|
| **Age (years)** | | |
| 15–25 | 158 | 42.5 |
| 26–35 | 175 | 47.0 |
| 36–49 | 39 | 10.5 |
| **Type of residence** | | |
| Rural | 131 | 35.2 |
| Peri urban | 116 | 31.2 |
| Urban | 125 | 33.6 |
| **Religion of Woman** | | |
| Christian | 313 | 84.2 |
| Islam | 12 | 3.2 |
| African Traditional Religion | 47 | 12.6 |
| **Marital status** | | |
| Married | 350 | 94.1 |
| Single | 22 | 5.9 |
| **Type of marriage** | | |
| Monogamy | 327 | 93.4 |
| Polygamy | 23 | 6.6 |
| **Woman education** | | |
| No formal Education | 47 | 12.6 |
| Basic Education | 167 | 44.9 |
| Secondary | 97 | 26.1 |
| Tertiary | 61 | 16.4 |
| **Occupation** | | |
| Unemployed | 165 | 44.3 |
| Formal Sector Workers | 55 | 14.8 |
| Informal Sector Workers | 152 | 40.9 |
| **Food Security Status** | | |
| Severe Food secure | 81 | 21.8 |
| Mild–Moderate | 133 | 35.9 |
| **Food Secure** | 158 | 42.5 |
| Type of Household toilet | | |
| Unimproved Facility | 241 | 64.8 |
| Improved Facility | 131 | 35.2 |
| **Parity** | | |
| 0 | 100 | 26.9 |
| 1 | 121 | 32.5 |
| 2 | 75 | 20.2 |
| ≥3 | 76 | 20.4 |
| **Health status** | | |
| Poor & Fair | 18 | 4.8 |
| Good | 133 | 35.8 |
| Very Good | 128 | 34.4 |
| Excellent | 93 | 25.0 |
| **Type of ANC Provider** | | |
| CHPS | 96 | 25.8 |
| Health Centre | 170 | 45.7 |

(*Continued*)

**Table 2.** (Continued)

| Variable | Frequency | Percentage (%) |
|---|---|---|
| Hospital | 44 | 11.8 |
| Regional Hospital | 62 | 16.7 |

(AOR = 2.25, 95%CI: 1.02–4.97, p = 0.044) and the odds became higher after adjusting for haemodilution for women registering in the 2nd trimester (AOR = 2.80, 95%CI: 1.18–6.68, p = 0.020). However, ANC registrants at private hospitals were less likely to be anaemic compared to those who registered at CHPS compounds (AOR = 0.32, 95%CI: 0.11–0.92), p = 0.035) and with more reduced odds after adjusting for haemodilution (AOR = 0.25,95%CI: 0.06–0.98), p = 0.047). Late (after 12th week of gestation) registrants were nearly two times more likely to be anaemic at booking compared to those registering in the 1st trimester (AOR = 1.87,95%CI: 1.17–2.98, p = 0.009). However, the association became statistically insignificant after adjusting for haemodilution (AOR = 0.68, 95%CI: 0.4 0–1.13, p = 0.05).

At 28 weeks of gestation, Christians were statistically significantly less likely to be anaemic compared to non-Christians (AOR = 0.34,95%CI: 0.31–0.89, p<0.05) but this relationship became statistically insignificant after adjusting for haemodilution. Women who attained tertiary education were statistically significantly less likely to be anaemic compared to women with no formal education (AOR = 0.09, 95%CI: 0.01–0.54, p = 0.009) but the significance of this association was not observed after adjusting for haemodilution (AOR = 0.18, 95%

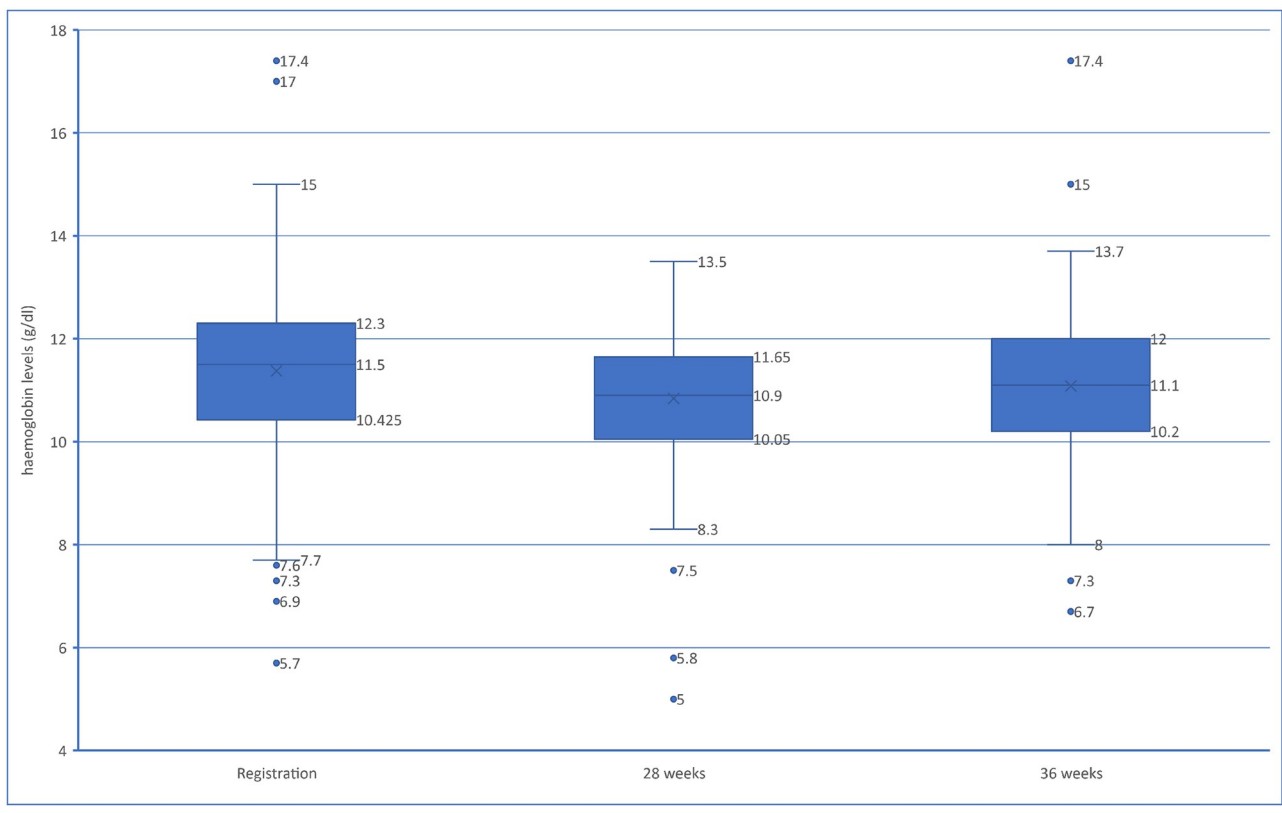

**Fig 1. Distribution of haemoglobin levels at the three stages of gestation.**

**Table 3. Prevalence of anaemia at the three stages of haemoglobin test among pregnant women.**

| Variable | Sample | Number anaemic | Prevalence (%) | 95% CI | |
|---|---|---|---|---|---|
| | | | | Lower | Upper |
| At Registration | 372 | 133 | 35.8 | 30.9 | 40.9 |
| HA* at Registration | 372 | 94 | 25.3 | 20.9 | 30.0 |
| At 28 weeks gestation | 192 | 102 | 53.1 | 45.8 | 60.3 |
| HA* at 28 weeks | 192 | 72 | 37.5 | 30.6 | 44.8 |
| At 36 weeks of gestation | 315 | 141 | 44.8 | 39.2 | 50.4 |

*HA = Haemodiluted adjusted Anaemia (10.5g/dl cut off for anaemia)

CI:0.03–1.15, p = 0.077. Women in the 5th wealth quintile were statistically significantly less likely to be anaemic compared to women in the 1st quintile (AOR = 0.27, 95%CI: 0.09–0.83, p = 0.022). The risk of anaemia at 28 weeks for wealthier household further reduced for the highest wealth quintile (AOR = 0.20, 95%CI 0.06–0.65, p = 0.008) and extended to cover women in the 4th quintile (AOR = 0.21, 95%CI: 0.06–0.75, p = 0.015) after haemodilution was adjusted for. Women in the middle reproductive age group (26–34 years) were less likely to be anaemic at 28 weeks (AOR = 0.46, 95%CI: 0.21–0.98 p = 0.044) and the odds further reduced after accounting for haemodilution(AOR = 0.33, 95%CI: 0.13–0.81, p = 0.001). Also, women booking after 1st trimester were over three times more likely to be anaemic after adjusting for haemodilution (AOR = 3.22 (95%CI: 1.42,7.29), p = 0.005). Also, after adjusting for haemodilution, women from urban residence were less likely to be anaemic at 28 weeks compared to those from rural residence (AOR = 0.31, 95%CI: 0.10–0.95, p = 0.041). Women whose spouses had basic education had four-fold increased odds of anaemia at 28 weeks (AOR = 4.00, 95%CI: 1.28–12.49, p<0.017) but the significance of this association was also lost after adjusting for haemodilution.

At 36 weeks, women in the middle reproductive age group (26–35 years) (AOR = 0.38, 95% CI: 0.21–0.68, p = 0.001) and in the oldest reproductive age group (35+ years) (AOR = 0.35, 95%CI: 0.13–0.90, p = 0.024) were statistically significantly less likely to be anaemic compared to women in the early reproductive age group (15–25 years). Also, those in the oldest reproductive age group (35+ years) were statistically significantly less likely to be anaemic compared to women in the early reproductive age group (15–25 years). Women from the richest households (5th wealth quintile) were also, statistically significantly less likely to be anaemic compared to those from the poorest households (1st wealth quintile) (AOR = 0.44, 95CI: 0.20–0.99, p = 0.049). Lastly, women registering late (after 1st trim) were statistically significantly less likely to be anaemic compared to those registering earlier (1st trim) (AOR = 1.72, 95%CI: 1.05, 2.84, p = 0.033).

## Discussion

This study assessed the prevalence of AIP and its correlates at three stages of hemoglobin assessment during ANC. The prevalence of AIP at ANC registration was 35.8%using a cut-off of 11.0g/dl and 25.3% using a cut-off of 10.5g/dl for women registering in their second trimester; at 28 weeks of gestation as 53.1% using a cut-off of 11g/dl and 37.5% when 10.5 haemoglobin cut-off was used; and at 36 weeks as 44.8%. The variations in AIP status among participants could be explained by a number of factors including maternal age, religion, education, household wealth, place of residence, type of healthcare facility, and timing of ANC

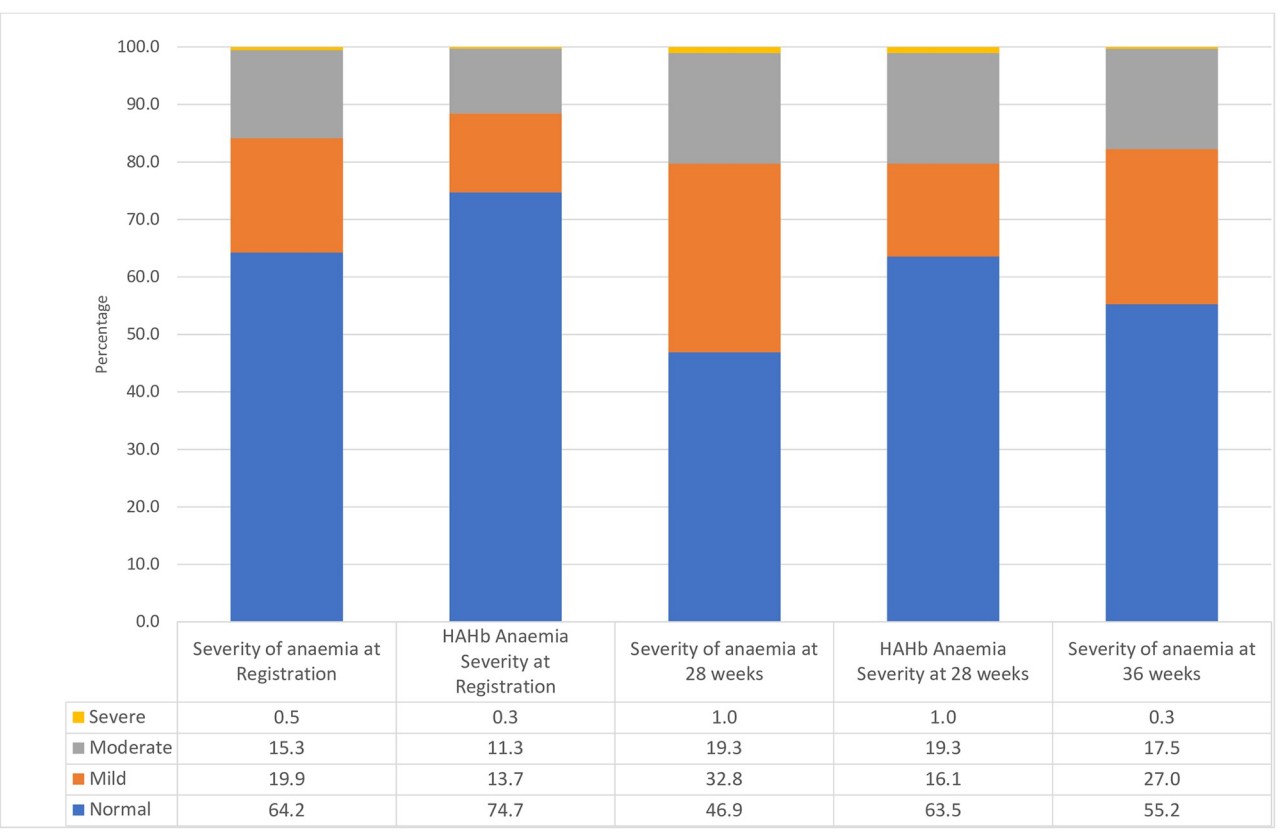

**Fig 2. Severity of anaemia status among pregnant women.**

registration, which were found to be statistically significantly associated with AIP at various stages of hemoglobin assessment.

Without adjustment for haemodilution, the observed AIP prevalence reported at ANC registration, was lower compared to similar studies both in Ghana [35, 48, 49] and in neighboring countries [54–56]. Also, the AIP prevalence at 36 weeks in this study varied compared to some studies in the northern and middle belts of Ghana [57, 58]. Contextual differences in study settings [6, 59–61] are more likely to account for these observed differences in AIP prevalence. While maternal healthcare is free in Ghana, the availability, access to, utilization and the quality of healthcare varies between the northern and middle belts of Ghana [62]. The interventions at ANC are expected to correct anaemia identified at registration and at 28 weeks of gestation and protect non-anaemic registrants from becoming anaemic. AIP prevalence should, therefore, be lesser in subsequent points of test especially at ≥36 weeks of gestation. This study however, illustrates worsening AIP prevalence after ANC registration. One study assessing AIP at booking and 36 weeks also reported a worsening situation [63]. On the contrary, a similar study in Ghana and two other studies elsewhere reported reductions in prevalence between registration and 36 weeks [45, 46, 49]. The study shows AIP is still of great concern, especially as it is one of the leading cause of maternal mortality [6, 64]. Efforts should therefore be strengthened to address AIP. Preconception nutrition and healthcare which will prevent or reduce AIP at registration are recommended.

In current practice, anaemia is diagnosed using a single cut off point of 11g/dl throughout pregnancy regardless of the time of assessment. As seen in this study, this practice presents a

**Table 4. Socio-economic factors associated with anaemia at the three stages of haemoglobin check during ANC.**

| Category | Registration | | HA at Registration | | 28 weeks | | HA at 28 weeks | | 36 weeks | |
|---|---|---|---|---|---|---|---|---|---|---|
| | AOR (95%CI) | P | AOR (95%CI) | P | AOR (95%CI) | P | AOR (95%CI) | P | AOR (95%CI) | P |
| Age | | | | | | | | | | |
| 15–25 | Ref | | Ref | | Ref | | Ref | | Ref | |
| 26–35 | 0.70(0.41 1.19) | 0.706 | 0.58(0.32, 1.04) | 0.070 | 0.46(0.21, 0.98) | 0.044 | 32.9(0.13, 0.81) | 0.016 | 0.38(0.21, 0.68) | 0.001 |
| 36–49 | 0.85(0.36, 2.01) | 0.646 | 0.90(0.35, 2.31) | 0.822 | 0.78(0.21, 2.97) | 0.716 | 0.67(0.16, 2.90) | 0.595 | 0.35(0.13, 0.90) | 0.024 |
| Type of residence | | | | | | | | | | |
| Rural | Ref | | Ref | | Ref | | Ref | | Ref | |
| Peri urban | 0.95(0.51, 1.77) | 0.877 | 0.65(0.32, 1.35) | 0.128 | 0.64(0.24, 1.79) | 0.379 | 0.56(0.19, 1.62) | 0.286 | 1.01(0.53, 1.92) | 0.983 |
| Urban | 0.78(0.40, 1.53) | 0.468 | 0.53(0.25, 1.13) | 0.102 | 0.48(0.17, 1.36) | 0.167 | 0.31(0.10, 0.95) | 0.041 | 0.86(0.43, 1.75) | 0.685 |
| Religion | | | | | | | | | | |
| Others | Ref | | Ref | | Ref | | Ref | | Ref | |
| Christianity | 0.53(0.28, 1.02) | 0.056 | 0.65(0.32, 1.35) | 0.249 | 0.34(0.31,0.89) | 0.028 | 0.49(0.18, 1.33) | 0.162 | 1.17(0.57, 2.39) | 0.677 |
| Educational status of women | | | | | | | | | | |
| None | Ref | | Ref | | Ref | | Ref | | Ref | |
| Basic | 0.80(0.38, 1.69) | 0.578 | 0.96(0.42, 2.20) | 0.925 | 0.74(0.24, 2.30) | 0.597 | 0.95(0.28, 3.201) | 0.929 | 0.65(0.29, 1.45) | 0.287 |
| Secondary | 0.58(0.23, 1.43) | 0.241 | 0.42(0.15, 1.18) | 0.101 | 0.55(0.15, 2.02) | 0.372 | 0.45(0.11, 1.81) | 0.265 | 0.62(0.24, 1.66) | 0.344 |
| Tertiary | 0.79(0.26, 2.41) | 0.677 | 0.69(0.19, 2.48) | 0.571 | 0.09(0.01, 0.54) | 0.009 | 0.18(0.03, 1.20) | 0.077 | 1.15(0.36, 3.71) | 0.811 |
| Occupational categories of women | | | | | | | | | | |
| Unemployed | Ref | | Ref | | Ref | | Ref | | Ref | |
| Formal | 0.69(0.29, 1.64) | 0.409 | 0.89(0.34, 2.32) | 0.807 | 2.23(0.49, 10.10) | 0.300 | 2.45(0.47, 12.63) | 0.285 | 0.57(0.23, 1.40) | 0.219 |
| Informal | 0.72(0.42, 1.23) | 0.227 | 0.76(0.42, 1.40) | 0.381 | 1.06(0.48, 2.31) | 0.891 | 0.93(0.39, 2.21) | 0.867 | 1.26(0.72, 2.21) | 0.428 |
| Educational statuses of spouses | | | | | | | | | | |
| None | Ref | | Ref | | Ref | | Ref | | Ref | |
| Basic | 0.83(0.42, 1.63) | 0.577 | 0.82(0.39, 1.74) | 0.604 | 4.00(1.28, 12.49) | 0.017 | 2.01(0.60, 6.70) | 0.255 | 0.78(0.38, 1.62) | 0.506 |
| Secondary | 0.98(0.43, 2.23) | 0.971 | 1.45(0.59, 3.50) | 0.425 | 3.04(0.90, 10.28) | 0.074 | 1.33(0.37, 4.79) | 0.654 | 0.35(0.14, 0.88) | 0.026 |
| Tertiary | 0.55(0.20, 1.52) | 0.252 | 0.84(0.27, 2.63) | 0.763 | 4.60(0.98, 21.47) | 0.052 | 1.16(0.24, 5.71) | 0.852 | 0.66(0.22, 1.94) | 0.450 |
| Occupational categories of spouses | | | | | | | | | | |
| Unemployed | Ref | | Ref | | Ref | | Ref | | Ref | |
| Formal | 2.60 (0.78, 8.61) | 0.118 | 1.08(0.29, 4.03) | 0.911 | 1.98(0.38, 10.39) | 0.419 | 6.05(0.79, 46.41) | 0.0083 | 3.80(0.86, 16.85) | 0.076 |
| Informal | 1.23(0.39, 3.89) | 0.724 | 1.00(0.28, 3.50) | 0.988 | 1.69(0.36, 7.86) | 0.503 | 1.77(0.26, 11.98) | 0.561 | 2.55(0.59, 10.93) | 0.208 |
| Quintile of household wealth index | | | | | | | | | | |
| 1 | Ref | | Ref | | Ref | | Ref | | Ref | |
| 2 | 0.86(0.41, 1.80) | 0.686 | 1.27(0.53, 3.07) | 0.590 | 0.96(0.29, 3.16) | 0.941 | 0.825(0.27, 2.70) | 0.778 | 0.82(0.38, 1.78) | 0.606 |
| 3 | 1.08(0.53, 2.23) | 0.827 | 2.22(0.97, 5.09) | 0.061 | 0.82(0.25, 2.63) | 0.732 | 0.78(0.24, 2.48) | 0.679 | 0.73(0.34, 1.56) | 0.416 |
| 4 | 1.04(0.50, 2.17) | 0.910 | 2.30(0.99, 5.35) | 0.053 | 0.42(0.13, 1.29) | 0.128 | 0.21(0.06, 0.75) | 0.015 | 1.04(0.48, 2.24) | 0.918 |
| 5 | 0.88(0.42, 1.84) | 0.727 | 1.35(0.56, 3.22) | 0.505 | 0.27(0.09, 0.83) | 0.022 | 0.20(0.06, 0.65) | 0.008 | 0.44(0.20, 0.99) | 0.049 |
| Household food security Status | | | | | | | | | | |
| Food Secure | Ref | | Ref | | Ref | | Ref | | Ref | |
| M–M | 0.89 (0.46, 1.73) | 0.336 | 0.98(0.52, 1.83) | 0.936 | 1.79(0.66, 4.81) | 0.488 | 0.88(0.37, 2.10) | 0.772 | 1.56(0.87, 2.80) | 0.140 |
| Severe FI | 1.17 (0.63, 2.19) | 0.723 | 0.76(0.36, 1.57) | 0.453 | 1.34(0.53, 3.37) | 0.251 | 0.41(0.13, 1.23) | 0.112 | 0.75(0.36, 1.59) | 0.457 |
| Level of Care Provider | | | | | | | | | | |
| CHPS | Ref | | Ref | | Ref | | Ref | | Ref | |
| Health Centre | 0.97(0.54, 1.74) | 0.926 | 1.00(0.53, 1.91) | 0.997 | 0.60 (0.21, 1.73) | 0.341 | 0.66(0.21, 2.12) | 0.483 | 1.08(0.57, 2.05) | 0.804 |
| Private Hosp. | 0.32(0.11, 0.92) | 0.035 | 0.25(0.06, 0.98) | 0.047 | 0.60(0.13, 2.82) | 0.521 | 1.35(0.25, 7.27) | 0.725 | 1.10(0.44, 2.78) | 0.838 |
| Regional Hosp. | 2.25(1.02, 4.97) | 0.044 | 2.80(1.18, 6.68) | 0.020 | 1.76(0.50, 6.21) | 0.381 | 3.50(0.90, 13.60) | 0.071 | 1.85(0.76, 4.50) | 0.175 |
| Trimester of Booking | | | | | | | | | | |
| 1st trimester | Ref | | Ref | | Ref | | Ref | | Ref | |

(*Continued*)

**Table 4.** (Continued)

| Category | Registration | | HA at Registration | | 28 weeks | | HA at 28 weeks | | 36 weeks | |
|---|---|---|---|---|---|---|---|---|---|---|
| | AOR (95%CI) | P | AOR (95%CI) | P | AOR (95%CI) | P | AOR (95%CI) | P | AOR (95%CI) | P |
| 2nd or 3rd trimester | 1.87(1.17, 2.98) | 0.009 | 0.68(0.40, 1.13) | 0.135 | 1.36(0.66, 2.80) | 0.400 | 3.22(1.42, 7.29) | 0.005 | 1.72(1.05, 2.84) | 0.033 |
| Constant | 1.13 | 1.00 | 1.02 | 0.980 | 2.38 | 0.231 | 0.72 | 0.731 | 0.69 | 0.707 |
| Observations | 372 | | 372 | | 192 | | 192 | | 315 | |
| Omnibus test $R^2$ (df) | 42.00(25) | 0.018 | 42.54(25) * | 0.016 | 39.83* (25) | 0.030 | 52.7851(25) | 0.001 | 40.88* (25) | 0.024 |
| Hosmer and Lemeshow $R^2$ (df) | 10.56(8) | 0.226 | 4.76(8) | 0.783 | 5.10(8) | 0.756 | 10.09(8) | 0.259 | 11.12(8) | 0.195 |
| -2 loglikelihood | 443.03 | | 378.02 | | 225.59 | | 199.13 | | 392.34 | |
| Cox & Snell $R^2$ | 0.11 | | 0.11 | | 0.19 | | 0.22 | | 0.12 | |
| Nagelkerke $R^2$ | 0.15 | | 0.16 | | 0.25 | | 0.30 | | 0.16 | |

FI = Food Insecurity HA = Haemodilution Adjusted $R^2$ = Chi Square

totally different pattern of anaemia prevalence along the course of pregnancy taken into account the haemodilution theory [65]. Without adjusting for haemodlition, AIP peaks at 28 weeks with a much higher prevalence. However, after adjusting for haemodilution, a rising straight-line curve pattern, demonstrating a continuous increasing prevalence from registration to 36 weeks, was observed. This could have some serious implication for planning for anaemia control and prevention in pregnancy. Without haemodilution, healthcare providers would expect the highest prevalence of AIP around 28 weeks and may place so much efforts on it. However, this could be deceptive and lead to poor prioritization of care. The adjustment changed the prevalence pattern which could affect AIP decision making Haemodilution is a natural physiological process in pregnancy that leads to decrease haemoglobin levels by 1–2g/dl but does not imply a disease state [22]. Indeed, there is an argument that haemoglobin above 14.30g/dl (43% haematocrit) may signal poor haemodilution which poses danger to mother and baby [65]. Hence, the argument for an adjustment of 0.5g/dl to be made for diagnosis at the 2nd trimester where haemodilution peaks is useful [24, 66]. Thus, there is the need to review the anaemia diagnosis at 28 weeks of gestation. Perhaps it is time for Ghana to adopt the cutoff point of 10.5g/dl as recommended by WHO [24, 66]. The different prevalence pattern of AIP as observed vis a vis haemodilution may also be a diagnosing error in anaemia research where all pregnant women regardless of their state of haemodilution are assessed using a common cutoff.

Age offered significant protection at 36 weeks, as older age groups had reduced risk of AIP. While this finding may be limited to women at 36 weeks of gestation, it has been reported in other studies of increased AIP risk for younger mothers [41, 56, 67–69] and protection for older mothers [70]. Also, the women in the middle age group had some protection against anaemia at 28 weeks when haemodilution is not taken into consideration. This also falls in line with some studies that found isolated risk for specific age groups [71, 72]. Also, the lack of association between anaemia at registration and age groups as observed in this study has also been reported in other studies [34, 35, 73].

The trimester of booking was a significant correlate of anaemia status throughout the three stages of assessment. The consistence was however achieved after adjusting for haemodilution at 28 weeks. This finding is similar to some assertions of increased risk of anaemia for registering after the 1st trimester of pregnancy [35, 43, 74] and echoes the report of increased risk with increasing trimesters of registration [75]. It is however, in variance with Ampiah et al.'s finding of reduced risk for registering in the 2nd trimester [50]. Several reasons may account for this.

Even though feeding issues such as appetite, nausea and vomiting are said to predominate in the 1st trimester [76], early booking enables early intervention, thereby enhancing nutrient intake. However, these likely determinants were not assessed in this study. The effects of these are also most likely to be felt in later trimesters than in 1st trimester, thereby increasing the risk in those times. Furthermore, early registration could increase the likelihood of more doses of iron and folic acid consumed thereby reducing the risk of anaemia even in later points of hae-moglobin check. This suggests the need to enable and encourage women to register early for ANC as a strategy for anaemia control and prevention in pregnancy.

The type of health care provider had a significant influence on anaemia status at only registration. Attending ANC at the private hospitals was associated with reduced risk of anaemia at registration but attending ANC at the regional hospital was associated with increased risk of anaemia at registration. It is important to note that more women (64%) registered early in the private hospitals than all other facilities. In practice, the cost of services at the private facilities are higher than public facilities in Ghana even when the client has NHIS. Women registering at private hospitals are therefore likely to be more committed to achieving good pregnancy outcomes or are "wealthy" women. Hence, they will most likely register early and comply with recommended behaviours. On the other hand, more (53.2%) women booked late at the Regional Hospital. Also, as a public facility and referral centre, clients booking there may be less likely to be committed and/or referred there due to previous risky (including anaemia risk) pregnancies. Hence, such women could have chosen the regional hospital because of pre-known anaemia issues before booking.

Residents of urban areas had some protection against anaemia at 28 weeks after adjusting for haemodilution. Previous studies have reported similar findings [11, 41, 77, 78]. Availability, accessibility and utilization of highly skilled and comprehensive ANC or anaemia-related care; and arguably, at a lower cost is an advantage to urban women over their rural counterparts. Certainly, travel expenses disrupts care access and utilization by pregnant women [79] and so will place rural women at the disadvantage side. For instance, NHIS accredited laboratories, pharmacies and drug stores are all located in the urban areas of the municipality. Ayoya et al. reported that inaccessibility to Iron and Folic Acid supplementation is far more common in rural areas than in urban settings [77]. In addition, women in urban areas are more likely to have higher (tertiary) education and be members of richest households which are both protective factors of anaemia at 28 weeks as reported in this study [80]. As such, they stand more empowered to carry out recommended anaemia prevention and control behaviours that offer protection.

Christianity offered statistically significant protection against anaemia at 28 weeks. The difference in anaemia in pregnancy risk determined by religion has also been reported in India [78]. Religion as a social determinant of health shapes the beliefs, norms, myths and conventions surrounding health and health-related behaviours [81]. These health-related behaviours could range from food and hygiene norms to health seeking ones that can impact on the incidence and recovery from anaemia. Though not assessed in this study, the religious norms regarding animal foods which are high bioavailable sources of protein, iron and other micro-nutrients needed for haemoglobin formation could account for this.

Similarly, educational status of spouses offered some protection against anemia at 36 weeks but only at the secondary level. Generally, education as a social determinant of health has been linked to higher nutrition and health literacy [81, 82] and higher economic fortunes [80, 81, 83] which directly impact on healthy behaviour leading to better health outcomes. Women with the highest level of education are therefore more likely to have the cognitive and affective ability to access, understand, appraise anaemia prevention and control information as well as the material resources to use that information to stay non-anaemic or quickly recover from

anaemia at 28 weeks or 36 weeks of pregnancy. Moreover, such women have good social standings and are therefore more likely to receive the needed attention. They are also more likely to be able to navigate effectively in the healthcare settings to demand care that meets their needs. This therefore calls for the education of the female child for better nutrition and anaemia outcomes in pregnancy.

Lastly, women in the first two wealthier household socio-economic groups as expected were protected against anaemia at 28 weeks and 36 weeks in this study. Previous studies have also documented similar findings [11, 78]. Wealth has been identified as a determinant of several health outcomes [81, 84]. While how wealth impact on anaemia status may be complex [81], the increased likelihood to have effective demand to anaemia prevention and control related commodities such as drugs, nutritious food and others is an advantage for women from wealthy households. This therefore may be the pathway to protection against anaemia in pregnancy.

### Strengths and limitations of the study

Firstly, this study analyzed the anaemia status of the same women at three critical stages of gestation. This permitted the exploration of progress of anaemia in pregnant women and it allowed for paired analysis to detect changes that occurred between points of haemoglobin assessment. Secondly, data on haemoglobin were collected via case records from three different source documents. This allowed for triangulation of information [85] and hence effectively addressed issues of recall bias and inaccuracies. The design of this study has possible limitations. Measurement of haemoglobin was done at various facilities using different Hb measuring machines of various qualities. This could influence haemoglobin values between cases and points of assessment. Thus, errors due to instrumentation were unavoidable [86]. In addition, the use of secondary data comes with its own demerits. One weakness is the failure to document care given to pregnant women. Data on other important variables such as gestational weight at ANC registration were not gathered and hence such variables were excluded from the analysis. This could have accounted for the nearly over 48% of the data on haemoglobin checked not available for 28 weeks. The above limitation therefore reduced the sample size for some aspects of the analysis.

### Conclusion

The prevalence of anaemia worsened from registration to 36 weeks when haemodilution is taken into account but peaked at 28 weeks if haemodilution is not considered. The study revealed that the correlates of AIP differed between registration, 28 weeks and 36 weeks of pregnancy. Late registration and the type of ANC provider determined AIP risk at registration. At 28 weeks, age groups, religion of mother, place of residence, trimester of registration, household wealth index and education influenced the risk of AIP. Additionally, the determinants of AIP at 36 weeks were household wealth index, age group, spousal education status and trimester of registration. We recommend a review of the haemoglobin cut-off for diagnosing anaemia in pregnancy. Behaviour change communication for the treatment of AIP should be tailored according to individual case scenarios as the socioeconomic determinants of AIP differ. Ghana's efforts to improve the educational status of girls should be sustained. It is also recommended that a larger prospective cohort study should be carried out to further complement the findings of this cross-sectional study.

### Author Contributions

**Conceptualization:** Donatus Nbonibe Abaane, Gilbert Abotisem Abiiro.

**Data curation:** Donatus Nbonibe Abaane.

**Formal analysis:** Donatus Nbonibe Abaane, Martin Nyaaba Adokiya, Gilbert Abotisem Abiiro.

**Investigation:** Donatus Nbonibe Abaane.

**Methodology:** Donatus Nbonibe Abaane, Gilbert Abotisem Abiiro.

**Project administration:** Donatus Nbonibe Abaane.

**Resources:** Donatus Nbonibe Abaane.

**Software:** Donatus Nbonibe Abaane.

**Supervision:** Gilbert Abotisem Abiiro.

**Validation:** Donatus Nbonibe Abaane, Martin Nyaaba Adokiya, Gilbert Abotisem Abiiro.

**Visualization:** Donatus Nbonibe Abaane, Martin Nyaaba Adokiya, Gilbert Abotisem Abiiro.

**Writing – original draft:** Donatus Nbonibe Abaane.

**Writing – review & editing:** Donatus Nbonibe Abaane, Martin Nyaaba Adokiya, Gilbert Abotisem Abiiro.

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
