## [Decision Letter · Decision Letter 0]

22 Dec 2022

PONE-D-22-01981Factors associated with anaemia in pregnancy: A retrospective cross-sectional study in northern GhanaPLOS ONE

Dear Dr. Abaane,

Thank you for submitting your manuscript to PLOS ONE. I sincerely apologise for the unusually delayed review timeframe.

Your manuscript has been assessed by one reviewer. After careful consideration of their comments, we feel that it has merit but does not fully meet PLOS ONE’s publication criteria as it currently stands. Therefore, we invite you to submit a revised version of the manuscript that addresses the points raised during the review process. In particular, please ensure that you have updated the introduction and discussion with references to all relevant literature on this topic, and include a thorough discussion of the new information that your study provides relative to the published literature.

Please note that we have only been able to secure a single reviewer to assess your manuscript. We are issuing a decision on your manuscript at this point to prevent further delays in the evaluation of your manuscript. Please be aware that the editor who handles your revised manuscript might find it necessary to invite additional reviewers to assess this work once the revised manuscript is submitted. However, we will aim to proceed on the basis of this single review if possible.

We look forward to receiving your revised manuscript.

Kind regards,

Emily Chenette

Editor in Chief

PLOS ONE

Journal Requirements:

Reviewers' comments:

Reviewer's Responses to Questions

**Comments to the Author**

1. Is the manuscript technically sound, and do the data support the conclusions?

Reviewer #1: Yes

2. Has the statistical analysis been performed appropriately and rigorously? 

Reviewer #1: No

3. Have the authors made all data underlying the findings in their manuscript fully available?

Reviewer #1: Yes

4. Is the manuscript presented in an intelligible fashion and written in standard English?

Reviewer #1: Yes

5. Review Comments to the Author

Reviewer #1: This paper reports the factors associated with anemia at first antenatal booking, 28 weeks of pregnancy, and 36 weeks of pregnancy in the Bolgatanga Municipality, Northern Region of Ghana. I have the following comments:

1) Line 1 (Title): The Bolgatanga Municiaplity is a small part of what we know as northern Ghana, which includes 5 regions, namely the Northern, Savanna, North-East, Upper East, and Upper West regions. Thus, authors should insert "Bolgatanga Municipality" in the title, so it reads: "Factors associated with anaemia in pregnancy: A retrospective cross sectional study in the Bolgatanga Municipality in northern Ghana".

2) Line 35: It would be helpful to include the mean (or median) gestational weight at registration.

3) Line 36-38: Authors' use of "after haemodilution" is confusing. Why not use something like "after WHO adjustment" or simply "using anemia cut off of 11.0 g/dl" versus "using anemia cut-off of 10.5 g/dl"?

4) Line 39-40: Authors should use was/were "associated with increased odds" or "associated with decreased odds" .....

5) Line 90-92: Readers do not believe that "only two studies done elsewhere" (Refs 46 and 47) have ever reported the hemoglobin concentration and anemia status of women at more than one time point during pregnancy?

6) Line 92-94: This (in addition to Line 90-92) is a poor description of study rationale and appears to suggests that authors are not familiar with the Hb/anemia literature of pregnant women in Ghana. I can give you at least one of such studies in Ghana:

Adu-Afarwuah S, Lartey A, Okronipa H et al. (2017) Impact of small-quantity lipid-based nutrient supplement on hemoglobin, iron status and biomarkers of inflammation in pregnant Ghanaian women. Maternal & Child Nutrition (2017), 13, e12262..

Was the primary aim of the study not to determine the factors associated with anemia at first antenatal booking, 28 weeks of pregnancy, and 36 weeks of pregnancy in the Bolgatanga Municipality?

7) Line 124-126: In line 110, "haemoglobin checked at registration was only about 52% at 36 weeks of gestation". So, given this "retrospective cross-sectional study", did authors randomly select participants from among those with Hb values at all 3 time points (first book, 28 wk, and 36 wk)?

8) Line 132-134: This is a "retrospective" study, so this part is confusing, i.e., "a structured questionnaire was used to collect data on household food security and additional sociodemographic characteristics including household assets, that were not captured in the medical records." Authors should describe clearly how the women were contacted for the survey. Were the hospital records reviewed before or after the survey? For a "retrospective cross-sectional study", were all participants enrolled after 36 weeks of pregnancy?

9) Line 142-143: At what time point (in terms of gestational age) were the women contacted? Were (some) women contacted before 36 wk of pregnancy? If so, how is this a "retrospective" study?

10) Line 155-156: Comparison of means at 3 time points: it is confusing how they used a dependent (paired) t-test here. Please provide more details on how the analyses were organized.

11) Line 158-159: Comparison of percentages at 3 time points. Please provide details of how the analyses were organized, eg, (1 v 2, 1 v 3, and 2 v 3?.

12) Line 160-162: Please describe how the predictors were selected for the final binary regression models.

13) Table 3: What are the P-values comparing?

14) Table 4 is very confusing:

a. Why did authors decide to separate non-anemic versus anemic women? Was this part of the objective, in addition to examining mean Hb concentrations at the 3 time points?

b. What are the "adjusted" and "unadjusted" Hb values? How was the adjustment done?

I see the comparisons were: Registration vs 28 wk, Registration vs 36 wk, and 28 wk v 36 wk. I'm sure readers this presentation boring; the table is long and difficult to follow (with the use of "adjusted" and "unadjusted" Hb). Maybe should consider using a line graph to present these data.

15) Table 5: How did authors select predictor variables for the logistic regression models? Did they simply throw all the background socioeconomic and food insecurity, etc., variables into the models? That is why this table is so big.

6. PLOS authors have the option to publish the peer review history of their article (what does this mean?). If published, this will include your full peer review and any attached files.

Reviewer #1: No

---

## [Author Response · Author response to Decision Letter 0]

18 Feb 2023

Dear Editor

A rebuttal letter that responds to each point raised by the academic editor and reviewer(s) have been uploaded as a separate file labeled 'Response to Reviewers.

A marked-up copy of the manuscript highlighting changes made to the original version has also been uploaded as a separate file labeled 'Revised Manuscript with Track Changes'.

An unmarked version of the revised paper without tracked changes has also been uploaded as a separate file labeled 'Manuscript'.

---

## [Decision Letter · Decision Letter 1]

11 May 2023

Factors associated with anaemia in pregnancy: A retrospective cross-sectional study in the Bolgatanga Municipality in northern Ghana

PONE-D-22-01981R1

Dear Dr. Donatus Nbonibe Abaane

We’re pleased to inform you that your manuscript has been judged scientifically suitable for publication and will be formally accepted for publication once it meets all outstanding technical requirements.

Kind regards,

Mpho Keetile, PhD

Academic Editor

PLOS ONE

Additional Editor Comments (optional):

Reviewers' comments:

Reviewer's Responses to Questions

**Comments to the Author**

1. If the authors have adequately addressed your comments raised in a previous round of review and you feel that this manuscript is now acceptable for publication, you may indicate that here to bypass the “Comments to the Author” section, enter your conflict of interest statement in the “Confidential to Editor” section, and submit your "Accept" recommendation.

Reviewer #1: All comments have been addressed

2. Is the manuscript technically sound, and do the data support the conclusions?

Reviewer #1: Yes

3. Has the statistical analysis been performed appropriately and rigorously? 

Reviewer #1: Yes

4. Have the authors made all data underlying the findings in their manuscript fully available?

Reviewer #1: Yes

5. Is the manuscript presented in an intelligible fashion and written in standard English?

Reviewer #1: Yes

6. Review Comments to the Author

Reviewer #1: (No Response)

7. PLOS authors have the option to publish the peer review history of their article (what does this mean?). If published, this will include your full peer review and any attached files.

Reviewer #1: No

---

## [Editor Report · Acceptance letter]

16 May 2023

PONE-D-22-01981R1 

Factors associated with anaemia in pregnancy: A retrospective cross-sectional study in the Bolgatanga Municipality, northern Ghana 

Dear Dr. Abaane:

I'm pleased to inform you that your manuscript has been deemed suitable for publication in PLOS ONE. Congratulations! Your manuscript is now with our production department. 

Kind regards, 

on behalf of

Dr. Mpho Keetile 

Academic Editor

PLOS ONE